# Algorithm for Determining Three Components of the Velocity Vector of Highly Maneuverable Aircraft

**Volodymyr Pavlikov [1,\*], Eduard Tserne [1,\*], Oleksii Odokiienko [1], Nataliia Sydorenko [1], Maksym Peretiatko [1], Olha Kosolapova [1], Ihor Prokofiiev [1], Andrii Humennyi [2] and Konstantin Belousov [3]**

[1] Aerospace Radio-Electronic Systems Department, National Aerospace University "Kharkiv Aviation Institute", 61070 Kharkiv, Ukraine
[2] National Aerospace University "Kharkiv Aviation Institute", 61070 Kharkiv, Ukraine
[3] Spacecraft, Measuring Systems and Telecommunications Department, Yuzhnoye SDO, 49000 Dnipro, Ukraine
[\*] Correspondence: v.pavlikov@khai.edu (V.P.); e.tserne@khai.edu (E.T.)

**Abstract:** We developed a signal processing algorithm to determine three components of the velocity vector of a highly maneuverable aircraft. We developed an equation of the distance from an aircraft to an underlying surface. This equation describes a general case of random spatial aircraft positions. Particularly, this equation considers distance changes according to an aircraft flight velocity variation. We also determined the relationship between radial velocity measured within the radiation pattern beam, the signal frequency Doppler shift, and the law of the range changing within the irradiated surface area. The models of the emitted and received signals were substantiated. The proposed equation of the received signal assumes that a reflection occurs not from a point object, but from a spatial area of an underlying surface. It fully corresponds to the real interaction process between an electromagnetic field and surface. The considered solution allowed us to synthesize the optimal algorithm to estimate the current range and three components $\{V_x, V_y, V_z\}$ of the aircraft's velocity vector $\vec{V}$. In accordance with the synthesized algorithm, we propose a radar structural diagram. The developed radar structural diagram consists of three channels for transmitting and receiving signals. This number of channels is necessary to estimate the full set of the velocity and altitude vector components. We studied several aircraft flight trajectories via simulations. We analyzed straight-line uniform flights; flights with changes in yaw, roll, and attack angles; vertical rises; and landings on a glide path and lining up with the correct yaw, pitch, and roll angles. The simulation results confirmed the correctness of the obtained solution.

**Keywords:** aircraft radio electronics; velocity measurement; height measurement; signal processing algorithm

## 1. Introduction

Autonomy is one of the most important characteristics of aviation systems. It refers to the ability to receive all necessary information about both a flight (e.g., coordinates in space, velocity, and flight altitude) and the detected surrounding objects with aviation equipment. Autonomy also helps pilots make appropriate decisions regarding aircraft flight control and solve the tasks assigned to them. It allows pilots to considerably expand the area of effective aviation applications. Pilots can usually use different autonomy systems and levels to solve different problems. However, engineers are trying to design multifunctional systems [1,2]. An important feature of such systems is the wide range of measured parameters and characteristics of the studied objects with minimum on-board equipment [3]. However, the implementation of multifunctional systems often requires the development of new, more complex operation algorithms [4,5]. Implementing such algorithms by increasing the computational performance of programmable logic devices while reducing their power consumption is possible [6].

To implement the autonomy of the aircraft, constantly obtaining information about the current parameters of its movement is necessary. To determine this, scholars have synthesized various separate radio systems for navigation and traffic control [7–9]. Among these radio systems, the presence of gauges of three components of the aircraft velocity and flight altitude is fundamental for autonomous aircraft systems [10]. Traditionally, two different systems measure these parameters: a Doppler radar measures the full speed and angle of attack, and a radio altimeter measures the true altitude of the aircraft. At the same time, current radio electronic components and high-speed processing systems open new possibilities for the design of multifunctional systems. Such systems will allow researchers to minimize the radar volume and weight and simultaneously reduce energy consumption. Besides the technical aspects, creating a new structure of the single signal processing system and a new method to estimate aircraft movement parameters and positions in space is also relevant. Scholars have paid particular attention to helicopters, which are characterized by a higher degree of freedom in movement than airplane-type vehicles (hovering, vertical flight, backward flight, low-speed flight, and so on). This imposes remarkable limitations on problem solutions concerning signal processing algorithm synthesis for such a radar operation [10].

We conducted scientific research and synthesized a signal-processing algorithm for an advance-functional radar for measuring the full vector $\{V_x, V_y, V_z\}$ of velocity $\overrightarrow{V}$ and altitude with [10,11]. We were able to perform such a synthesis because of the achievement of the theory of the statistical optimization of radio engineering systems and the availability of modern computer systems to measure aircraft motion parameters in quasireal time [12–14].

## 2. Materials and Methods

*2.1. Geometry of the Problem. The Equation of the Distance to the Underlying Surface for a Maneuvering Aircraft*

To measure the velocity and altitude of a helicopter-type aircraft, developing a new radar with special orientation of the antennas and their radiation patterns is necessary. The radiation pattern beam must not be directed vertically down, but at some angle relative to the nadir direction. In this way, scholars will avoid zero or near zero values of the Doppler frequency shift [15]. Thus, when developing a new algorithm for the operation of a multifunctional radar, the position of the rays is assumed to already be fixed and rigidly related to the direction of the longitudinal axis of the aircraft. Figure 1 shows the primary geometry of the stated problem and describes the key parameters of a radiation pattern beam position in space.

In Figure 1, the transmitting $A_1$ and receiving $A_2$ antennas are assumed to be located in the center of the coordinate system $x'\,y'\,z'$, which is related to some point of the aircraft. $\Delta\theta_{pa}$ is the beam width of the radiation pattern. The parameters $D'_1$ and $D'_2$ describe the shapes of the antenna apertures. The radius vectors $\overrightarrow{r}'_1$ and $\overrightarrow{r}'_2$ characterize the distance from the phase center of the antenna to any point within its aperture. The angles $\mu$, $\varphi$, $\rho$, $\eta$ are used to determine the position of the radiation pattern in space. The radiation pattern irradiates a certain area of the underlying surface located in the coordinate system $x\,y\,z'$. Several main parameters characterize the range from the antenna system to the irradiated area. The first parameter is the distance $R$ between the phase center of the transmitting antenna and the center of the underlying surface of the irradiated area. The range to the nearest $R_{min}$ and farthest $R_{max}$ point of the irradiated area and the current range $R_{fl}$ to an arbitrary point $P$ within this area are also separately determined. Separately distinguishing a point within the area of equal distance, which is conventionally shown within the irradiated area of the surface, is not possible.

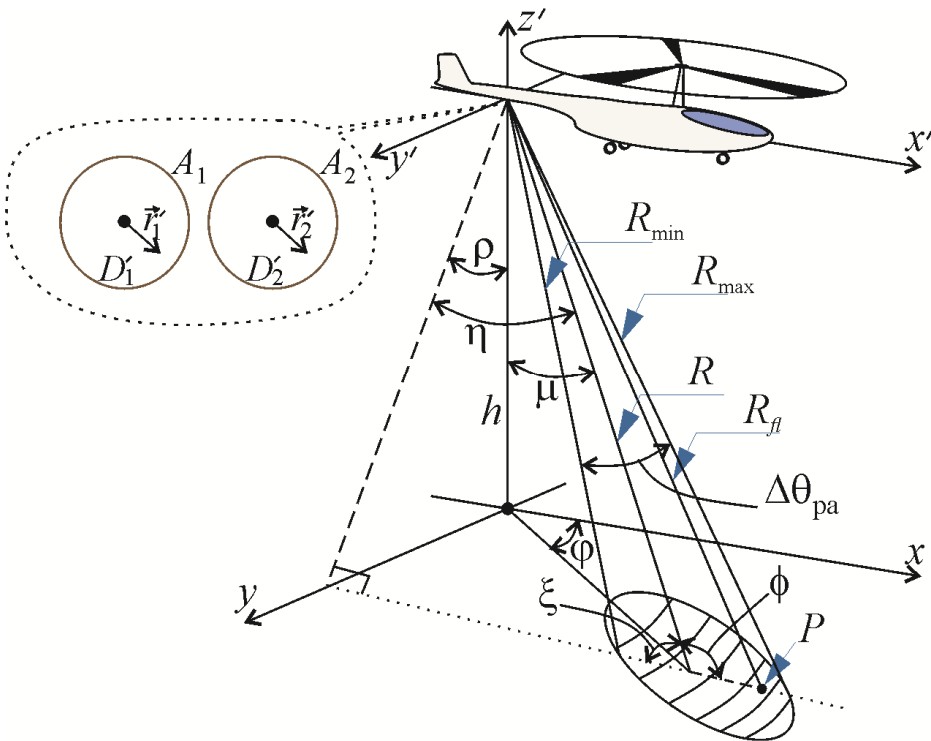

**Figure 1.** Primary geometry of the problem with marked physical parameters and geometric relationships for one radiation pattern beam.

A helicopter is a highly maneuverable aircraft [16]. Rapid changes in the helicopter position lead to essential variations in the antenna's radiation pattern direction in space. Considering this, the radiation pattern spot usually "slides" on the underlying surface along very complex trajectories. We used the geometry shown in Figure 2 to calculate the range with any changes in the yaw, pitch, and roll angles.

In Figure 2, the coordinate system 0xyz is related to the underlying surface. In this coordinate system, a helicopter is at a certain altitude $h'$ in the center of the $0'x'y'z'$ coordinate system. This coordinate system is the initial one and corresponds to the case when the aircraft does not perform any maneuvers, i.e., the yaw, roll, and pitch angles are equal to zero. In this case, the radiation pattern direction coincides with the line $R'$, which determines the current distance to the central irradiated point $M'$ on the underlying surface x0y. The angle $\eta$ to the axis $0'x'$ and the angle $\rho$ to the axis $0'y'$ characterize the line $R'$ position in space. The angle $\mu'$ between $R'$ and the nadir direction and the angle $\varphi'$ between the projection $R'$ onto the underlying surface and the axis 0x are also used for calculations. When a yaw angle $\alpha'$ appears in an aircraft movement, the transition from the $0'x'y'z'$ coordinate system to $0x''y''z''$ occurs. At the same time, direction $R''$ focuses the radiation pattern on point $M''$. All changes that occur during the appearance of the yaw angle are marked with a superscript $\cdot''$. When yaw $\alpha'$ and pitch $\theta''$ angles appear in the helicopter position, the transitions from the $0'x'y'z'$ coordinate system to $0x'''y'''z'''$ and to the variables with the superscript $\cdot'''$ occur. In the case of the presence of yaw $\alpha'$, pitch $\theta''$, and roll $\chi'''$ angles, the coordinate system changes to $0x^{IV}y^{IV}z^{IV}$. To understand coordinate transforms and an R angular position evaluation more clearly, we introduced a conditional sphere with the center at point $0'$ on the geometry in Figure 2. Straight lines $R^{(\cdots)}$ cross the sphere at the points $K'$, $K''$, $K'''$, or $K^{IV}$ depending on the current angles of the aircraft's position.

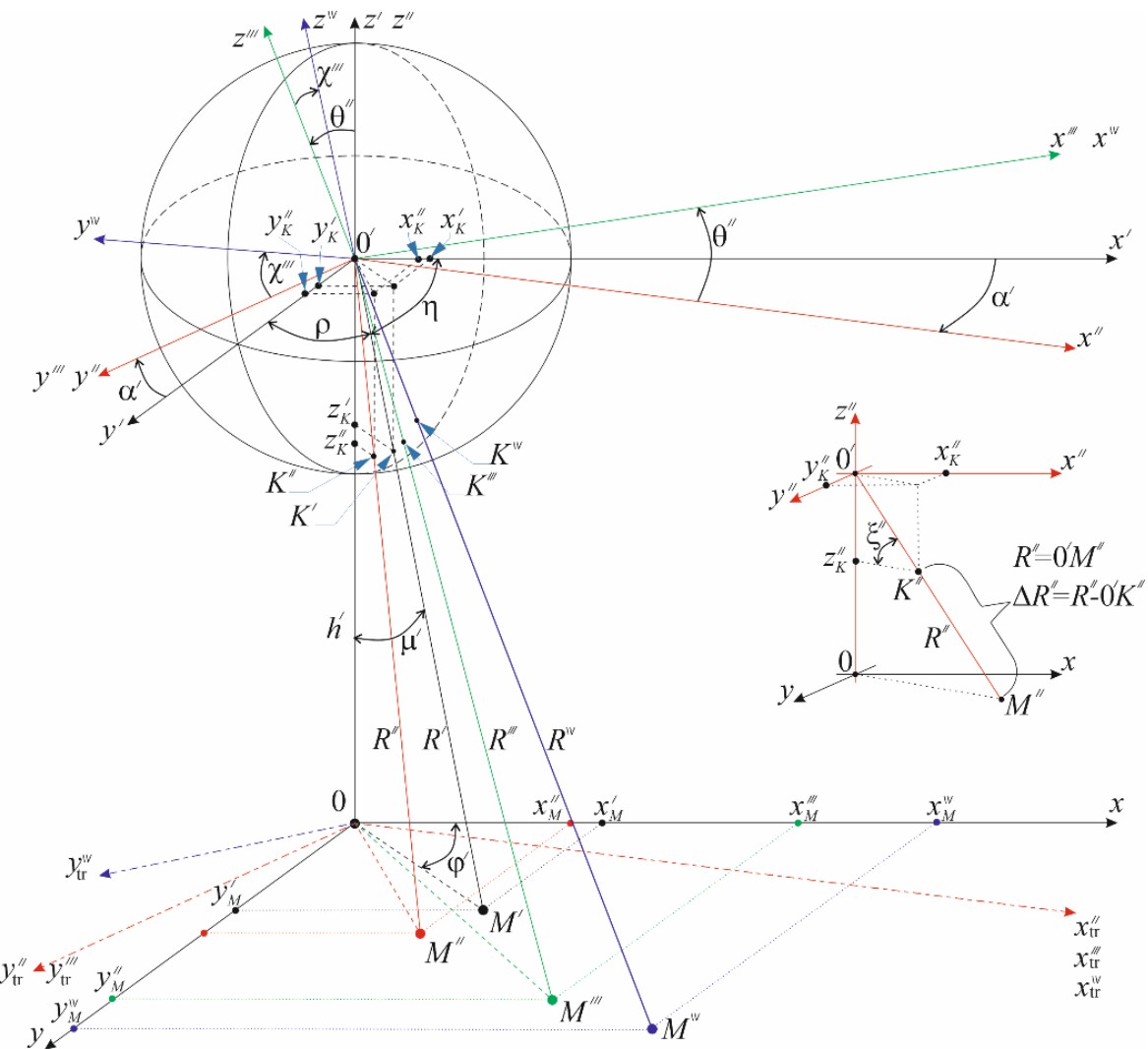

**Figure 2.** The geometry of the problem, which considers the presence of yaw, pitch, and roll angles in the aircraft movement.

Considering the geometry in Figure 2, possible changes in the yaw $\alpha'$, pitch $\theta''$, and roll $\chi'''$ angles in the equation for the range R calculation can be written in the following form [10]:

$$R(\alpha', \theta'', \chi''', h, \mu', \varphi', t) = R\left(t, \vec{r}\right)$$

$$= \frac{h(t)\left[\begin{array}{l}\left(\begin{array}{l}h(t)tg(\mu')\cos(\varphi')\cos(\alpha'(t))\cos(\theta''(t)) - h(t)tg(\mu')\sin(\varphi')\sin(\alpha'(t))\cos(\theta''(t)) \\ -h'(t)\sin(\theta''(t))\end{array}\right)^2 \\ +\left(\begin{array}{l}h(t)tg(\mu')\cos(\varphi')\sin(\alpha'(t))\cos(\chi'''(t)) + h(t)tg(\mu')\sin(\varphi')\cos(\alpha'(t))\cos(\chi'''(t)) \\ +h(t)tg(\mu')\cos(\varphi')\cos(\alpha'(t))\sin(\theta''(t))\sin(\chi'''(t)) \\ -h(t)tg(\mu')\sin(\varphi')\sin(\alpha'(t))\sin(\theta''(t))\sin(\chi'''(t)) + h(t)\cos(\theta''(t))\sin(\chi'''(t))\end{array}\right)^2 \\ +\left(\begin{array}{l}h(t)tg(\mu')\cos(\varphi')\sin(\alpha'(t))\sin(\chi'''(t)) + h(t)tg(\mu')\sin(\varphi')\cos(\alpha'(t))\sin(\chi'''(t)) \\ -h(t)tg(\mu')\cos(\varphi')\cos(\alpha'(t))\sin(\theta''(t))\cos(\chi'''(t)) \\ +h(t)tg(\mu')\sin(\varphi')\sin(\alpha'(t))\sin(\theta''(t))\cos(\chi'''(t)) - h(t)\cos(\theta''(t))\cos(\chi'''(t))\end{array}\right)^2\end{array}\right]^{\frac{1}{2}}}{\left(\begin{array}{l}h(t)tg(\mu')\cos(\varphi')\sin(\alpha'(t))\sin(\chi'''(t)) + h(t)tg(\mu')\sin(\varphi')\cos(\alpha'(t))\sin(\chi'''(t)) \\ -h(t)tg(\mu')\cos(\varphi')\cos(\alpha'(t))\sin(\theta''(t))\cos(\chi'''(t)) \\ +h(t)tg(\mu')\sin(\varphi')\sin(\alpha'(t))\sin(\theta''(t))\cos(\chi'''(t)) - h(t)\cos(\theta''(t))\cos(\chi'''(t))\end{array}\right)}.$$

(1)

In Equation (1), we assume that, during the flight, the height and angles of the yaw, pitch, and roll can change (that is, they are functions of time).

### 2.2. Models of Transmitted and Received Signals. Observation Equation

In this section, we formulate the requirements of the transmitted signal. To measure the flight velocity and altitude, the signal must have an ambiguity function that provides a high resolution in distance and speed. Complex signals usually have such characteristics. For our task, choosing a signal with a linear frequency modulation is reasonable [17,18]:

$$s(t) = A(t) \text{Re} \exp\left(j2\pi\left(f_0 t + \frac{\alpha t^2}{2}\right)\right), \tag{2}$$

where $A(t)$ is the signal envelop; $f_0$ is the emitted signal frequency; $\alpha = (F_{max} - F_{min})T^{-1}$; $T$ is the pulse duration; and $F_{max}$ and $F_{min}$ are the maximum and minimum frequencies in the spectrum of operating frequencies, respectively.

After radiation, the reflection from the underlying surface and registration by the receiving antenna signal (2) has the following form:

$$s_i(t) = \text{Re} \int_{D_i} \left|\dot{G}\left(\vec{r}\right)\right|^2 \dot{F}\left(\vec{r}\right) A\left(t - t_d\left(t, \vec{r}\right)\right) \exp\left(j2\pi\left[f_0\left(t - t_d\left(t, \vec{r}\right)\right) + 0.5\alpha\left(t - t_d\left(t, \vec{r}\right)\right)^2\right]\right) d\vec{r} \tag{3}$$

where the integration occurs over the irradiated antenna radiation pattern $\dot{G}\left(\vec{r}\right)$ area $D_i$ of the underlay surface (we assume that the radiation patterns of the transmission and receiving antennas are the same); $t_d\left(t, \vec{r}\right)$ is the signal delay time, which occurs as a result of its propagation from the antenna to the underlying surface elements and in the reverse direction; $\dot{F}\left(\vec{r}\right) = \left|\dot{F}\left(\vec{r}\right)\right| \exp\left(j\xi\left(\vec{r}\right)\right)$ is the complex reflection coefficient of the underlying surface; $\xi\left(\vec{r}\right)$ is the random phase offset that occurs when the signal is reflected from the underlying surface; and $t$ is the current time.

The delay time in Equation (2) is determined according to the following equation:

$$t_d\left(t, \vec{r}\right) = 2R_{fl}\left(t, \vec{r}\right)c^{-1} \tag{4}$$

where $R_{fl}\left(t, \vec{r}\right)$ is the current range related to the velocity of the aircraft. $R_{fl}\left(t, \vec{r}\right)$ is analytically presented in the following form:

$$R_{fl}\left(t, \vec{r}\right) = \left(\begin{array}{c} R^2\left(t, \vec{r}\right) + \left(R\left(t, \vec{r}\right)\dfrac{\sin\left(\frac{\Delta\theta_{pa}}{2}\right)}{\sin\left(\frac{\pi}{2} - a\cos\left(\frac{h}{R\left(t, \vec{r}\right)\cos\left(a\tan\left(\frac{a(t)}{b(t)}\right)\right)}\right) - \frac{\Delta\theta_{pa}}{2}\right)} - Vt\right)^2 \\ -2R\left(t, \vec{r}\right)\left(R\left(t, \vec{r}\right)\dfrac{\sin\left(\frac{\Delta\theta_{pa}}{2}\right)}{\sin\left(\frac{\pi}{2} - a\cos\left(\frac{h}{R\left(t, \vec{r}\right)\cos\left(a\tan\left(\frac{a(t)}{b(t)}\right)\right)}\right) - \frac{\Delta\theta_{pa}}{2}\right)} - Vt\right)\cos(\phi), \end{array}\right)^{\frac{1}{2}}$$

where $\Delta\theta_{pa}$ is the beam width of the radiation pattern, and

$$a(t) = h'(t)\text{tg}(\mu')\sin(\phi' + \alpha'(t))\cos(\chi'''(t)) + h'(t)\text{tg}(\mu')\cos(\phi' + \alpha'(t))\sin(\theta''(t))\sin(\chi'''(t))$$
$$+h'(t)\cos(\theta''(t))\sin(\chi'''(t)),$$
$$b(t) = h'(t)\text{tg}(\mu')\sin(\phi' + \alpha'(t))\sin(\chi'''(t)) - h'(t)\text{tg}(\mu')\cos(\phi' + \alpha'(t))\sin(\theta''(t))\cos(\chi'''(t))$$
$$-h'(t)\cos(\theta''(t))\cos(\chi'''(t)).$$

The observation equation (at the receiver input) is written as an additive mixture of the received signal and the internal noise of the receiver:

$$u_i(t) = s_i(t) + n_i(t), \tag{5}$$

where $n_i(t)$ is white Gaussian noise with a power spectral density of $0.5N_0$. The index i specifies multichannel reception with several receivers. A simulation will justify the number of channels.

## 3. Results

### 3.1. Signal Processing Algorithm Synthesis. Simulation Results

We will solve the stated problem using the maximum likelihood method. To do this, we wrote the likelihood functional in the following form [19,20]:

$$p\left(u_i(t) \middle| R_{fl,i}\left(t, \vec{r}\right)\right) = k \exp\left\{-\frac{1}{N_0}\int_0^T \left(u_i(t) - s_i\left(t, R_{fl,i}\left(t, \vec{r}\right)\right)\right)^2 dt\right\}, \tag{6}$$

where k is some variable that does not depend on the parameter being estimated.

To define the new signal processing algorithm, we wrote the likelihood equation in the following form:

$$\frac{\delta \ln p\left(u_i(t) \middle| R_{fl,i}\left(t, \vec{r}\right)\right)}{\delta R_{fl,i}\left(\vec{r}\right)} = \frac{\delta k}{\delta R_{fl,i}\left(\vec{r}\right)} - \frac{1}{N_0}\frac{\delta}{\delta R_{fl,i}\left(\vec{r}\right)}\int_0^T \left(u_i(t) - s_i\left(t, R_{fl,i}\left(t, \vec{r}\right)\right)\right)^2 dt = 0. \tag{7}$$

The estimated parameter $R_{fl,i}\left(\vec{r}\right)$ is related to the flight velocity. This relationship can be established by estimating the Doppler frequency from the exponent argument in Equation (3) considering Equation (4):

$$f_D = -\frac{2}{c}\frac{dR_{fl}\left(t, \vec{r}\right)}{dt}(f_0 + \alpha t) - \alpha\frac{2R_{fl}\left(t, \vec{r}\right)}{c}\left(1 - \frac{2}{c}\frac{dR_{fl}\left(t, \vec{r}\right)}{dt}\right).$$

and using a formal description of the Doppler frequency of a signal with linear frequency modulation:

$$f_D = \frac{2V_r\left(t, \vec{r}\right)}{c}(f_0 + \alpha t).$$

By equating the right-hand sides of the equations for $f_D$ and neglecting unimpactful terms, we obtain the following relationship between velocity and the current range:

$$V_r\left(t, \vec{r}\right) = -\frac{dR_{fl}\left(t, \vec{r}\right)}{dt}, \tag{8}$$

which has a clear physical meaning.

By solving the likelihood Equation (7), we obtain the following equation:

$$\int_0^T u_i(t)\frac{\delta}{\delta R_{fl,i}\left(\vec{r}\right)}s_i\left(t, R_{fl,i}\left(t, \vec{r}\right)\right)dt = \int_0^T s_i\left(t, R_{fl,i}\left(t, \vec{r}\right)\right)\frac{\delta}{\delta R_{fl,i}\left(\vec{r}\right)}s_i\left(t, R_{fl,i}\left(t, \vec{r}\right)\right)dt, \tag{9}$$

where the left part is the signal processing algorithm and the right part is the result of averaging the radar effect, which implements the signal processing algorithm. Here, the variational derivative of the signal can be represented as

$$\frac{\delta}{\delta R_{fl,i}\left(\overrightarrow{r}\right)} s_i\left(t, R_{fl,i}\left(t, \overrightarrow{r}\right)\right)$$

$$= \text{Re} \int_{D_i} \dot{F}\left(\overrightarrow{r}\right)\left|\dot{G}\left(\overrightarrow{r}\right)\right|^2 \left[\frac{\delta}{\delta R_{fl,i}\left(\overrightarrow{r}\right)} A\left(t - \frac{2R_{fl,i}\left(t,\overrightarrow{r}\right)}{c}\right)\right]$$

$$\times \exp\left(j2\pi\left[f_0\left(t - \frac{2R_{fl,i}\left(t,\overrightarrow{r}\right)}{c}\right) + \frac{\alpha\left(t-2R_{fl,i}\left(t,\overrightarrow{r}\right)c^{-1}\right)^2}{2}\right]\right) d\overrightarrow{r}$$

$$+ \text{Re}(-j)2\pi\dot{F}\left(\overrightarrow{r}\right)\left|\dot{G}\left(\overrightarrow{r}\right)\right|^2 A\left(t - \frac{2R_{fl,i}\left(t,\overrightarrow{r}\right)}{c}\right)\left\{\frac{2f_0}{c} + \frac{2\alpha t}{c} - \frac{4\alpha R_{fl,i}\left(t,\overrightarrow{r}\right)}{c^2}\right\}$$

$$\times \exp\left(j2\pi\left[f_0\left(t - \frac{2R_{fl,i}\left(t,\overrightarrow{r}\right)}{c}\right) + \frac{\alpha\left(t-2R_{fl,i}\left(t,\overrightarrow{r}\right)c^{-1}\right)^2}{2}\right]\right).$$

Having solved Equation (9), we can estimate the range $\hat{R}_{fl}\left(t, \overrightarrow{r}\right)$. Further, according to Equation (7) and the geometry shown in Figure 1, we obtained the radial velocity $V_r\left(t, \overrightarrow{r}\right)$ and the absolute velocity of the helicopter. The absolute velocity has the following form:

$$V_{hel}\left(t, \overrightarrow{r}\right) = \frac{\frac{d}{dt}R_{fl}\left(t, \overrightarrow{r}\right)}{\cos\left(\frac{\pi}{2} - \text{acos}\left(\frac{h(t)}{R\left(t,\overrightarrow{r}\right)\cos\left(\text{atan}\left(\frac{a(t)}{b(t)}\right)\right)}\right)\right)}. \tag{10}$$

The velocity Equation (9) does not show the direction, only its absolute value. Estimating all of the helicopter's velocity $\left(V_x, V_y, V_z\right)$ components with one antenna radiation pattern is difficult, as shown in Figure 1. Therefore, let us consider three rays, as shown in Figure 3, and perform a corresponding simulation.

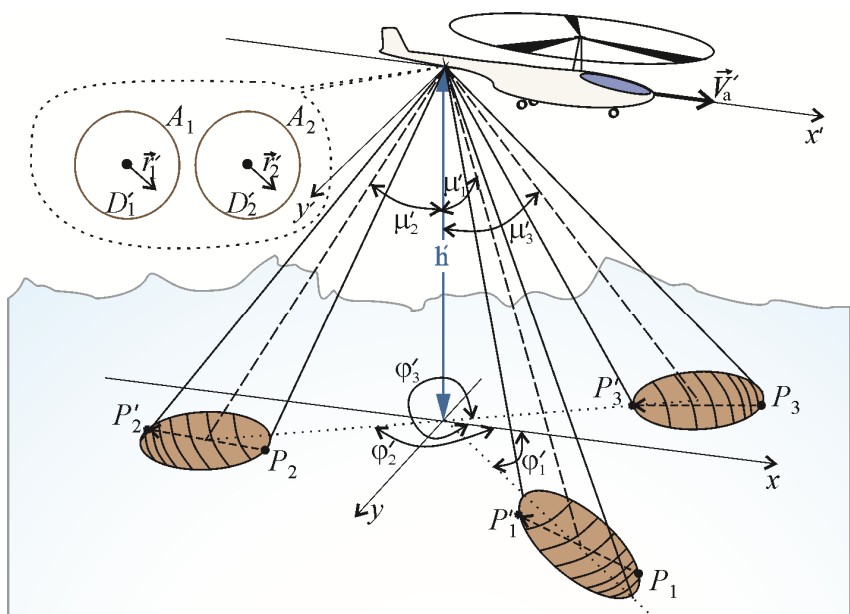

**Figure 3.** Geometry of the problem with three beams for the three components of the helicopter's velocity vector and altitude.

Figures 1 and 3 show a different number of radiation patterns. In Figure 3, we assumed that the apertures of the transmitting $A_1$ and receiving $A_2$ antennas formed three radiation

pattern beams in different directions. All of the beams are characterized by the parameters introduced in Figure 1. The subscript under each variable indicates which radiation pattern is under consideration. The angle between the beams is always fixed to $2\mu'$ and, to simplify the calculations, satisfying the condition $\mu'_1 = \mu'_2 = \mu'_3 = \mu'$ is advisable.

In the case of three beams, the helicopter velocity is measured according to Equation (10) in every beam. At the same time, the two components of the velocity vector can be determined as follows:

$$\begin{aligned} V_x(t) &= \frac{V_1(t) - V_2(t)}{2}, \\ V_y(t) &= \frac{V_1(t) - V_3(t)}{2}, \end{aligned} \tag{11}$$

where the velocity subscript in the right part indicates the number of beams according to Figure 3.

Now, let us determine the equation for determining the flight height and the third velocity vector component. Based on the geometry (Figure 3), the following formula can be written to determine a helicopter's flight height:

$$h(t) = \frac{R_2(\cdot, t)R_3(\cdot, t)}{\sqrt{R_2^2(\cdot, t) + R_3^2(\cdot, t) - 2R_2(\cdot, t)R_3(\cdot, t)\cos(2\mu')}}. \tag{12}$$

The third component of the velocity vector can be found in the following form:

$$V_z(t) = \frac{dh(t)}{dt}. \tag{13}$$

The beams can be placed in any way, but to measure low velocities, placing them mirror-like relative to the longitudinal and transverse axis is advisable, as shown in Figure 3. With this arrangement, doubling (see the numerators in Equation (11)) each component of the velocity vector while taking the measurements is possible and allows one to determine low velocities. We can prove this with simulations, the results of which are shown in Figures 4–7. For the first simulation, we utilized the following initial data: the law of height change was $h(t) = 1000 + 2t$; the glide angle was $\alpha'(t) = 0$; the angle of attack was $\theta''(t) = 0$; the pitch angle was $\chi'''(t) = 0$; and the velocity components were $V_x = 10\,\text{m/s}$ and $V_y = 0\,\text{m/s}$. Figure 4 shows the changes in the current range for each of the beams. Figure 5 describes the measured velocities along each of the beams. Figure 6 depicts three velocity components and Figure 7 shows an analysis of the height estimation.

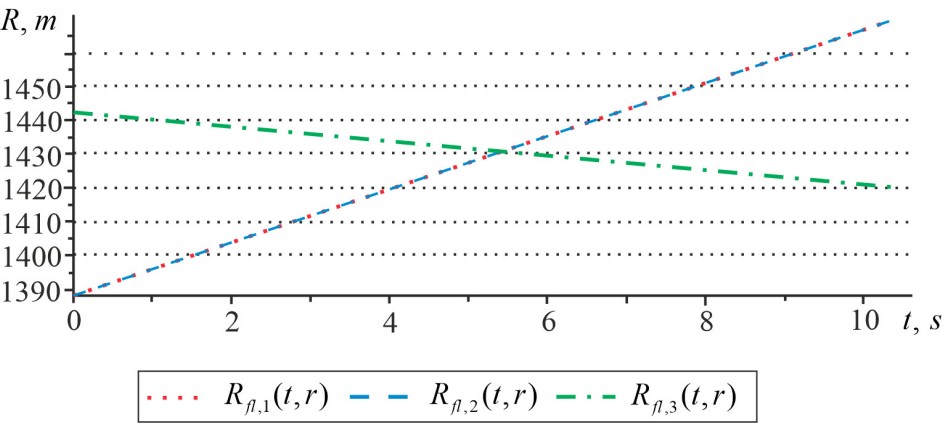

**Figure 4.** Variation in current distances along each of the beams (first simulation).

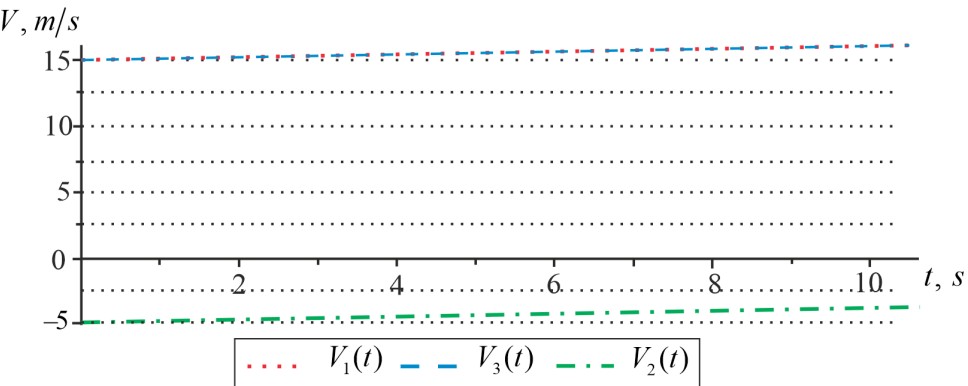

**Figure 5.** Measured absolute velocities along each of the beams (first simulation).

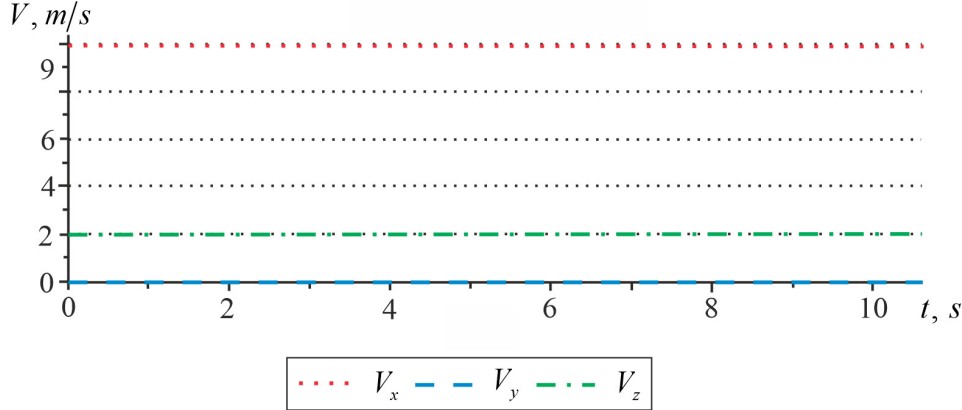

**Figure 6.** Estimates of the velocity vector components (first simulation).

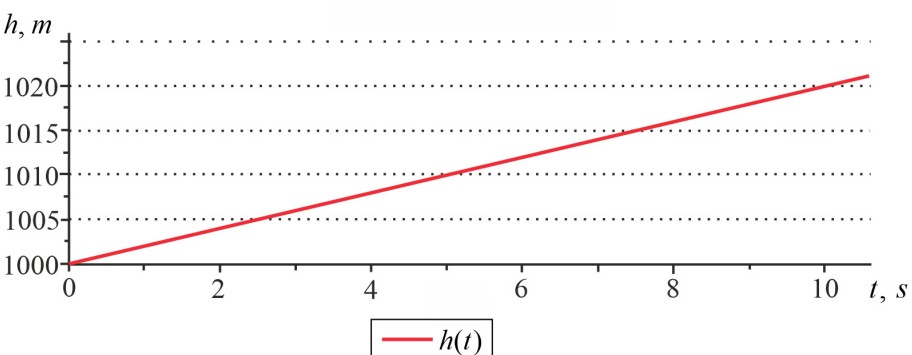

**Figure 7.** Height estimation (first simulation).

In the case of the flight height uniformly increasing, the absolute values of the velocity along each of the beams (Figure 5) did not correspond to the absolute velocity value. However, by analyzing Figure 6, one can see that the estimation of the velocity vector components during uniform movement was close to the true values, and the absolute error was less than 1.5%. We performed a range estimation according to Equation (12). The obtained results also corresponded to the stated initial data in the simulation.

Considering more complex helicopter flight trajectories when the yaw, attack, and pitch angles differ from zero is of interest. For this simulation, we used the following initial data: the height change law was $h(t) = 1000 + 2t$; the functional dependencies of the glide angle were $\alpha'(t) = (2t)°$; the attack angle was $\theta''(t) = (0.2t)°$; the pitch angle was $\beta'''(t) = 5°$; and the velocity vector modulus was $V = 10.6\,\text{m/s}$. Figures 8–11 depict the obtained graphs, which are similar to the results shown in Figures 4–7.

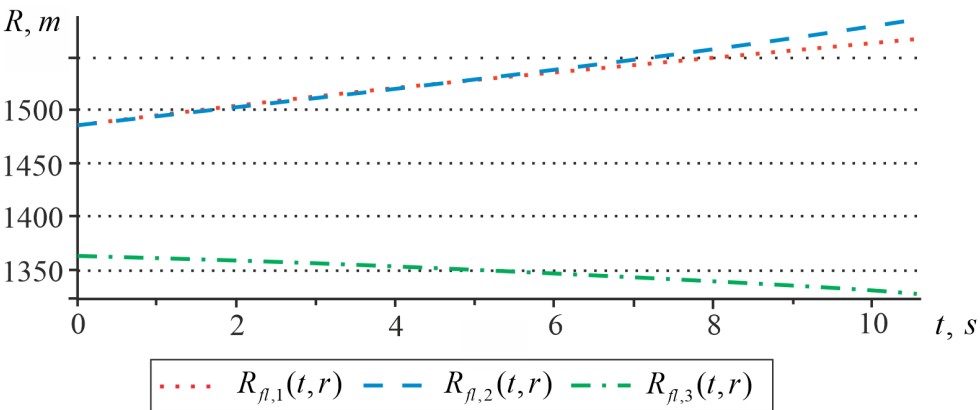

**Figure 8.** Measured change in current distances along each of the beams (second simulation).

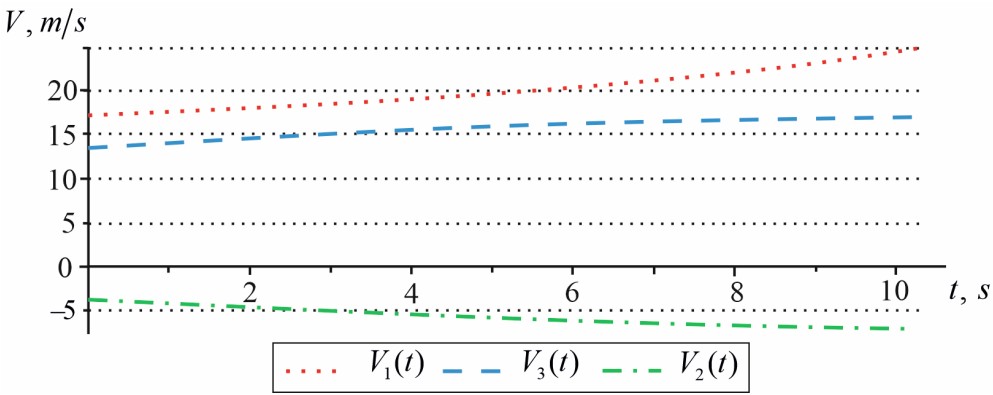

**Figure 9.** Measured absolute velocities along each of the beams (second simulation).

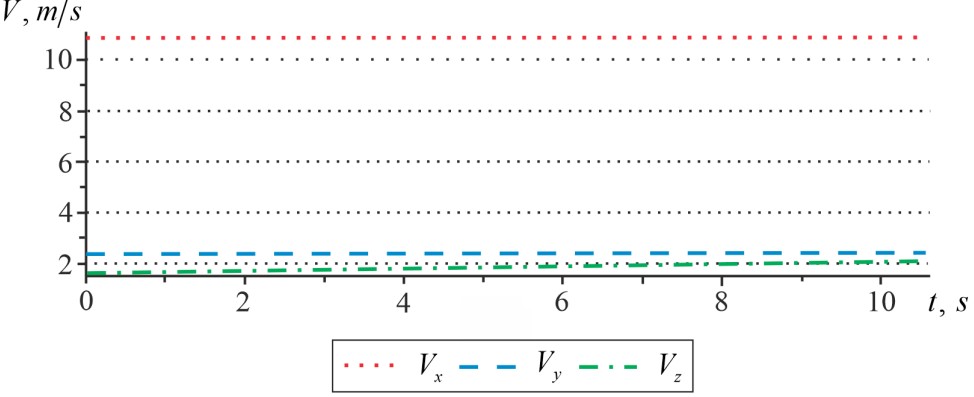

**Figure 10.** Estimates of the velocity vector components (second simulation).

In the presence of changes in the yaw, attack, and pitch angles during movement, the components of the velocity vector had more complex dependencies than was observed during uniform movement. This was because the ranges along each beam nonlinearly changed.

Considering the case of a vertical takeoff of a helicopter ($h(t) = V_z t$, $V_z = 3\,\mathrm{m/s}$, $t \geq 0$, $V_x = V_y = 0$). Figure 12 shows the estimated height and Figure 13 shows the estimates of the three velocity components.

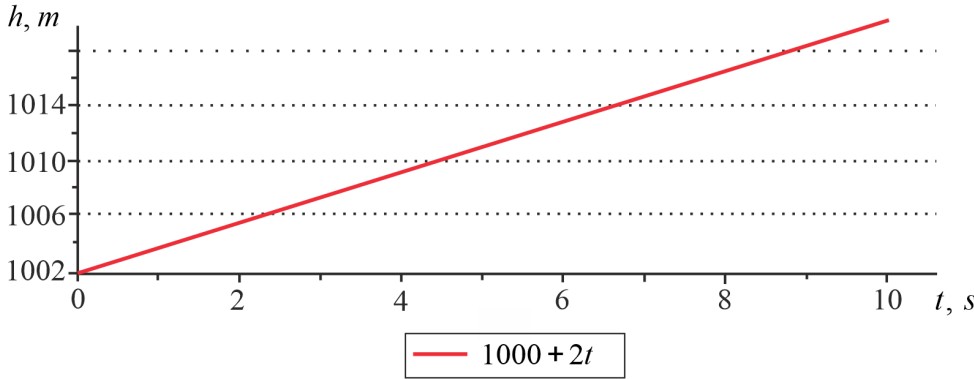

**Figure 11.** Height estimation (second simulation).

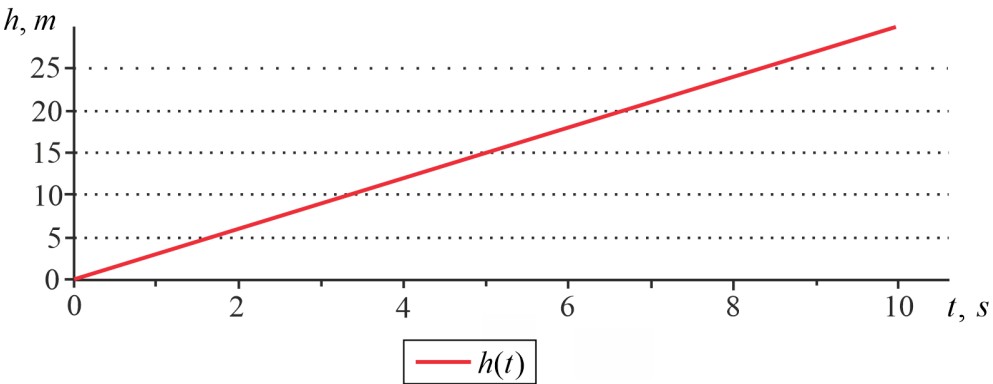

**Figure 12.** Estimation of height during vertical takeoff.

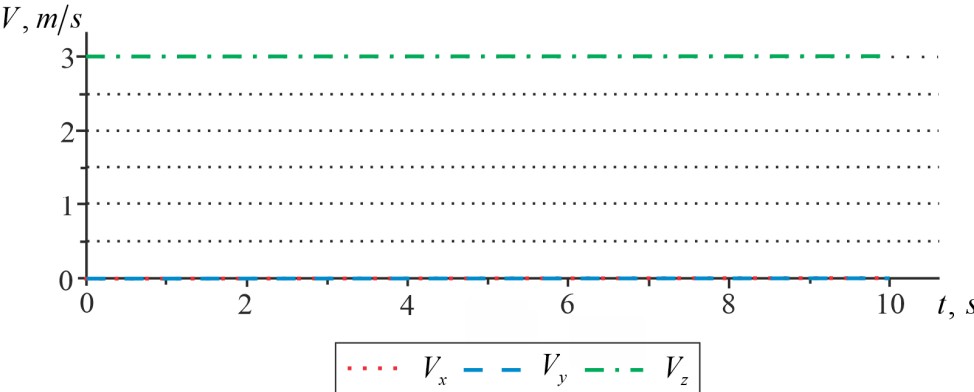

**Figure 13.** Estimates of the components of the velocity vector during vertical takeoff.

Now, we consider the landing option [21]. Figure 14 shows the simulated changes in the yaw, roll, and attack angles while landing. The height of the aircraft linearly decreased according to the law $h(t) = 50 - 5t$, as shown in Figure 15. The modulus of the velocity vector at the beginning of the simulation was 110 m/s. Figures 16 and 17 show the variation in distances along each of the rays and the changing components of the velocity vector.

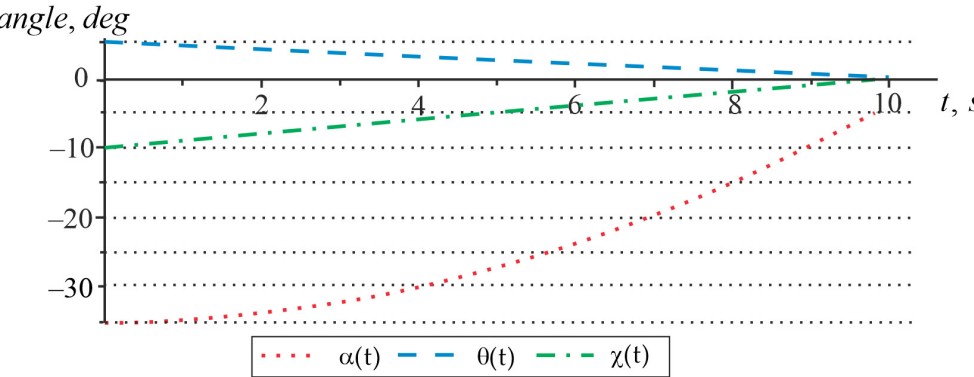

**Figure 14.** Changes in the angles of sliding, roll, and attack while a helicopter is landing.

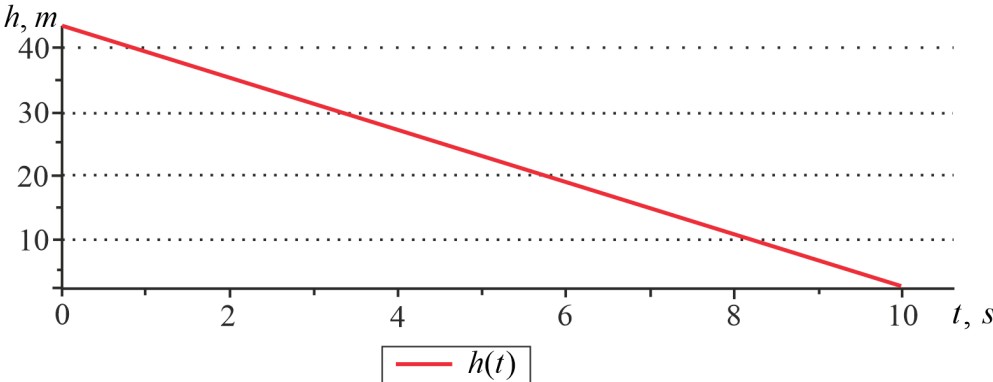

**Figure 15.** The height while a helicopter is landing.

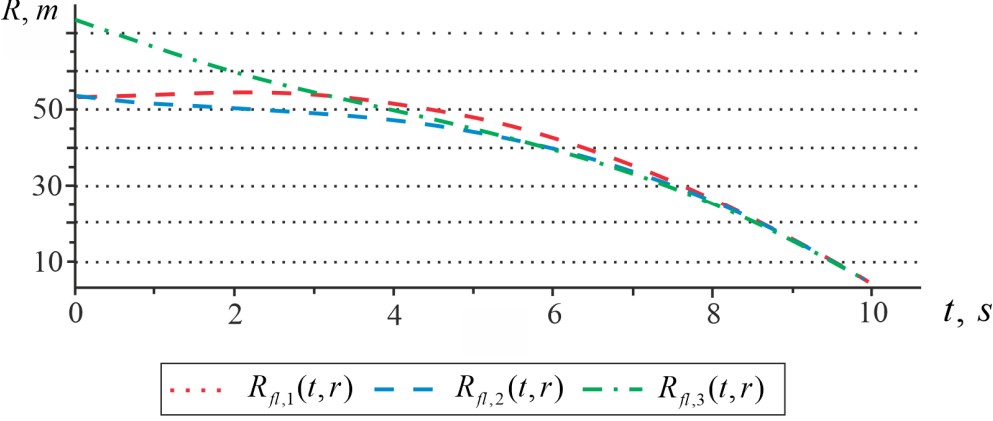

**Figure 16.** Calculated changes in distances along each of the beams while a helicopter is landing.

The obtained simulation results confirmed the adequacy of the performance of the derived equations and algorithms.

### 3.2. Structural Diagram of the Radar

We developed a structural diagram (Figure 18) of the radar by measuring a full vector of the velocity and flight height components according to the calculations.

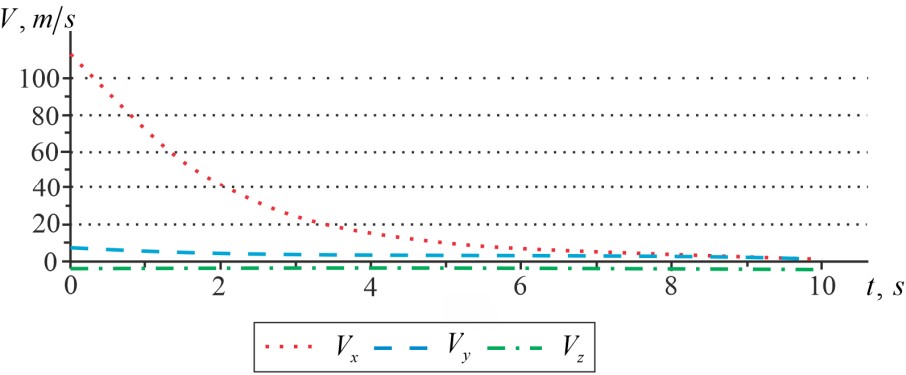

**Figure 17.** Estimates of the velocity vector components while a helicopter is landing.

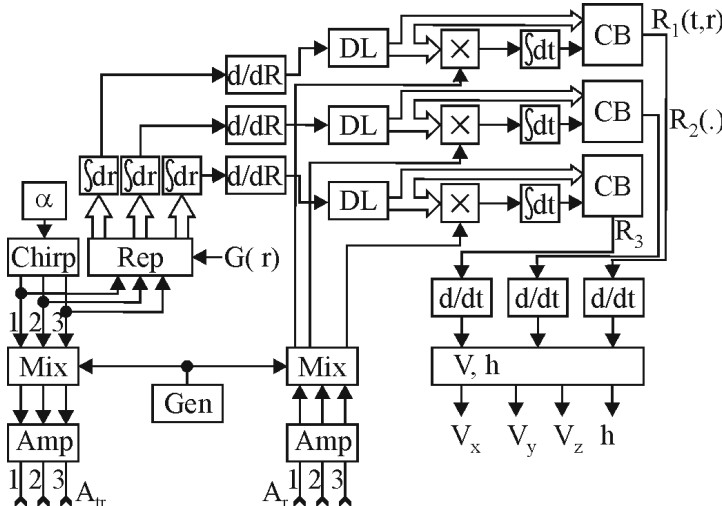

**Figure 18.** Structural diagram of the radar for the velocity vector components and altitude measurement.

The scheme has three transmitting and three receiving channels. It works as follows: The transmitter is represented by a signal generator with linear frequency modulation (chirp block with a setting block), mixers (mix block), and amplifiers (amp block). We assumed that three different signals were formed (three beams are enough to solve the problem), and we separated them in terms of frequency to exclude the possibility of channel mixing. The signals were transmitted through antennas $A_{tr}$. Antennas $A_r$ received the signals reflected by the underlying surface. After amplification in the amp block, the signals were transferred to an intermediate frequency in the mix block. Further, the signals were multiplied with the derivatives of the reference signals in the «×» blocks and then sent to the integrators. Derivatives of the reference signals entered the second input of the multipliers. Blocks of repeaters (Rep) were involved during the formation of the reference signals, which resulted in the reflection of the signal from a spatially extended area, integrators on spatial coordinates, blocks of variational derivatives, d/dR calculations, and delay lines (DLs). Then, from the outputs of the integrators, the processes were sent to the comparison block (CB), and distance estimates were formed at the outputs as a function of spatial and temporal coordinates. These range estimates passed through the differentiation blocks were converted into radial velocities (see Equation (8)) along each of the radiation patter beams and were sent to the block to calculate the components of the velocity vector and flight height according to Equations (11)–(13).

## 4. Discussion

One of the promising directions for modern aircraft development is increasing their autonomy, which requires the simultaneous real-time monitoring of many aircraft pa-

rameters and the use of a considerable number of different sensors and systems. At the same time, aircrafts have a remarkable limitation regarding the payload, which can be installed on board without deteriorating the tactical and technical characteristics of the machine. Therefore, developing multifunctional on-board systems that can simultaneously monitor several parameters and characteristics of the aircraft using a minimum amount of equipment is an urgent task. A possible direction for the creation of such systems is the development of new and advanced signal-processing algorithms. We obtained algorithms to calculate the full vector of velocities $\{V_x, V_y, V_z\}$ and flight altitude of an aircraft. These algorithms can be implemented in one multifunctional system.

We obtained the algorithm in Equation (9) as a result of the solution to likelihood Equation (7), considering the relationship between the velocity and the current range Equation (8). This algorithm allows one to calculate the velocity of an aircraft based on the measured range $\hat{R}_{fl}\left(t, \vec{r}\right)$. At the same time, in the presence of only one beam of the antenna radiation pattern, the obtained algorithm does not allow one to separately determine the aircraft velocity components $\{V_x, V_y, V_z\}$. Therefore, we created a transition to the geometry of the problem depicted in Figure 3, which involves the simultaneous use of three radiation patterns. The simultaneous use of three beams allowed us to directly calculate the components of the velocity along the y and x axis according to Equation (10), as well as the current flight height according to Equation (11). We calculated the third component of the velocity based on the height according to Equation (12).

We confirmed the general efficiency of the proposed algorithms with the simulation. Scholars should pay special attention to the calculated distances and velocities along each of the radiation pattern beams. Thus, during the second simulation, with a linear change in the glide $\alpha'(t)$ and attack $\theta''(t)$ angles, as well as height $h(t)$, we observed nonlinear changes in distances and velocities, as depicted in Figures 8 and 9. This fully corresponded to the real nonlinear range change dependence with a linear change in the vertical angle at which the radar probed the surface. Figures 5, 9 and 17 highlight the velocity $V_2(t)$, which had a negative value. This can be explained by the fact that the second radiation pattern beam on the geometry in Figure 3 was directed to the opposite side in relation to the aircraft movement direction. That is, the Doppler frequency behind this beam had a negative value, and it led to a negative measured velocity. The same effect could occur in real velocity measurement radars, and this also confirmed the correctness of the obtained results.

Based on the calculations and geometry of the problem shown in Figure 3, we developed a structural diagram of the on-board meter of the velocity and height vector components. Considering modern achievements in the field of radio element bases, the proposed scheme can be fully implemented. At the same time, when implementing such a system, paying attention to prospective frequency ranges, which are currently in the millimeter wave range, is advisable [22,23]. The implementation of an on-board system in this range in the future will allow scholars to remarkably reduce the weight and dimensions of the result system, which is fundamental for use on an aircraft, and the ability to measure the current height of the carrier will allow researchers to replace classic on-board radio altimeters.

We plan to continue this research in several directions in the future. First, we will obtain marginal errors when using the obtained algorithms for the velocity vector and height estimation. Additionally, we are working to create a radar that implements the proposed algorithms. As a result, we plan to determine the technical requirements for the hardware and signal processing speed and to obtain the first practical results of the velocity vector and flight altitude estimation on a real helicopter.

## 5. Conclusions

We explored the problem of synthesizing a signal-processing algorithm to estimate a current beam range, three components $\{V_x, V_y, V_z\}$ of a helicopter's velocity vector, and flight height. As a result, we solved several issues. We derived the equation of the distance to the underlying surface along the radiation pattern beam. Its general form allowed

us to analyze the distance for any carrier position above the underlying surface. This is an important result because modern helicopters have many diverse flight modes from hovering in place to flying in the reverse direction. For the first time, we propose the idea of calculating aircraft velocity vector components by measuring the distances along the beams. The results of numerical modeling showed that three transmitting and receiving channels were enough to solve the velocity vector component and altitude problem. We formulated the requirements for the signal type selection. We propose the use of a waveform with linear frequency modulations that allows one to obtain high-resolution distance and velocity measurements. We obtained equations and a structural diagram of a multifunctional radar for the flight height and three velocity vector components.

**Author Contributions:** Conceptualization, V.P., I.P. and O.O.; methodology, V.P. and O.O.; software, E.T. and M.P.; validation, A.H., K.B. and O.K.; formal analysis, V.P. and A.H.; investigation, O.O. and I.P.; resources, E.T.; data curation, N.S.; writing—original draft preparation, V.P. and N.S.; writing—review and editing, O.K. and K.B.; visualization, M.P.; supervision, A.H.; project administration, V.P.; funding acquisition, K.B. All authors have read and agreed to the published version of the manuscript.

**Funding:** This work was funded by the Ministry of Education and Science of Ukraine, and the state registration numbers of the projects are 0122U200469 and 0121U109598.

**Institutional Review Board Statement:** Not applicable.

**Informed Consent Statement:** Not applicable.

**Data Availability Statement:** Not applicable.

**Conflicts of Interest:** The authors declare no conflict of interest.

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
