# Peer review of "Algorithm for Determining Three Components of the Velocity Vector of Highly Maneuverable Aircraft"

_computation, doi:10.3390/computation11020035_

Round 1
Reviewer 1 Report
Dear authors, please check for correct use of the english language. Especially, the use of plural and singular forms for verbs. Currently, the paper is in my opinion not sufficient for publication.
Author Response
Dear reviewer!
Thank you for our article reviewing and for pointing out the shortcomings.
The paper has been reviced in accordance with your comment:
Pont 1: Dear authors, please check for correct use of the english language. Especially, the use of plural and singular forms for verbs.
Respons 1: We have carried out additional work to correct errors in the use of the English language.
Reviewer 2 Report
Dear authors,
I have read your document carefully. I found your proposal interesting. Along the text there are a few issues that you require to address it properly.
Attached there is a document that shows a few suggestions.
The International Units System is used in here, so please use it accordingly.
Please send the references at the end of the sentence.
Figure 1. It is a quite interesting image, please in the caption explain each one of the parameters that you used in here.
Figure 2, is quite magnificent and also, with lots and lots of info to explain. Your caption is quite simple. You need to improve it order to understand the set of equations that you are using such as Eq. (1), otherwise, it is only a fancy and useless image, fancy nevertheless.
Equation (1), you are using it as a monstruosity that can be highly improved. Why are you using a square root in here, if every section is squared, what are you aiming to analyse in here? in here, h(t) is also within the each of the equation terms, why it is h^3(t) within the root?
Please double check Eq.(2).
Instead of formula, please use equation or Eq.
Figure 14 & 15, how did you manage to have such angles?
What are the requirements in hardware that you algorithm require as well as which is the processing speed?
What is the power consumption and the error that you can get at high and low speeds?

Author Response
Dear reviewer!
Thank you for our article reviewing and and for your suggestions..
The paper has been reviced in accordance with your comment:
Point 1: The International Units System is used in here, so please use it accordingly.
Response 1: In the paper, the units have been formatted in accordance with the SI standard
Point 2: Please send the references at the end of the sentence.
Response 2: All references have been moved to the end of the sentences.
Point 3: Figure 1. It is a quite interesting image, please in the caption explain each one of the parameters that you used in here.
Response 3: The description of Figure 1 is improved in the main text.
Point 4: Figure 2, is quite magnificent and also, with lots and lots of info to explain. Your caption is quite simple. You need to improve it order to understand the set of equations that you are using such as Eq. (1), otherwise, it is only a fancy and useless image, fancy nevertheless.
Response 4: The description of Figure 2 has been completely changed in the main text. More information about the variables and transitions introduced on it is provided .
Point 5: Equation (1), you are using it as a monstruosity that can be highly improved. Why are you using a square root in here, if every section is squared, what are you aiming to analyse in here? in here, h(t) is also within the each of the equation terms, why it is h^3(t) within the root?
Response 5: Unfortunately, we cannot directly take out the individual squared brackets from the root, since their sum is calculated. The value h(t) has been initially located behind the root in the calculations. When writing the final version of Eq. (1), it has been decided not to include it under the root for the convenience of further calculations. Also we have changed the form of this formula due to incorrect display of the root.
Point 6: Please double check Eq.(2).
Response 6: Description of Eq. (2) has been corrected.
Point 7: Instead of formula, please use equation or Eq.
Response 7: References to equations have been changed in accordance with this proposal.
Point 8: Figure 14 & 15, how did you manage to have such angles?
Response 8: The description has been changed for this simulation. The angles indicated in Figure 14 are the initial data for the simulation. Such behavior of angles while landin was chosen in consultation with a pilots. The distances shown in Figure 15 are the result of Eq.1 permotming.
Point 9: What are the requirements in hardware that you algorithm require as well as which is the processing speed?
Response 9: We have just begun the practical implementation of the radar and cannot precisely define these requirements. They will be determined during further work. This information has been added to the discussion section.
Point 10: What is the power consumption and the error that you can get at high and low speeds?
Response 10: At the moment, we are calculating the marginal measurement errors while using the proposed algorithms. This information will be published in future articles. This information has been added to the discussion section. The power consumption will be determined during the implementation of the radar. At the simulation stage, it is difficult to determine this parameter.
Point 11: Suggestions in the document.
Response 11: In accordance with the suggestions in the document, additional changes have been made to the paper. Some paragraphs have been rewritten, additional information has been added to the Reference section.